# Spatial Distribution of Black Carbon Concentrations in the Atmosphere of the North Atlantic and the European Sector of the Arctic Ocean

Sergey M. Sakerin [1,*], Dmitry M. Kabanov [1,*], Vladimir M. Kopeikin [2], Ivan A. Kruglinsky [1], Alexander N. Novigatsky [3], Viktor V. Pol'kin [1], Vladimir P. Shevchenko [3] and Yuri S. Turchinovich [1]

[1] V.E. Zuev Institute of Atmospheric Optics, Siberian Branch, Russian Academy of Sciences, Academician Zuev Square 1, 634021 Tomsk, Russia; iak@iao.ru (I.A.K.); victor@iao.ru (V.V.P.); tus@iao.ru (Y.S.T.)

[2] A.M. Obukhov Institute of Atmospheric Physics, Russian Academy of Sciences, Pyzhevsky Pereulok 3, 119017 Moscow, Russia; kopeikin@ifaran.ru

[3] Shirshov Institute of Oceanology, Russian Academy of Sciences, Nakhimovsky Prospect 36, 117997 Moscow, Russia; novigatsky@ocean.ru (A.N.N.); vshevch@ocean.ru (V.P.S.)

* Correspondence: sms@iao.ru (S.M.S.); dkab@iao.ru (D.M.K.)

**Abstract:** We discuss the measurements of black carbon concentrations in the composition of atmospheric aerosol over the seas of the North Atlantic and European sector of the Arctic Ocean (21 expeditions in 2007–2020). The black carbon concentrations were measured by an aethalometer and filter method. The comparison of the two variants of the measurements of the black carbon concentrations showed that the data acceptably agreed and can be used jointly. It is noted that the spatial distribution of black carbon over the ocean is formed under the influence of outflows of air masses from the direction of continents, where the main sources of emission of absorbing aerosol are concentrated. We analyzed the statistical characteristics of black carbon concentrations in five marine regions, differing by the outflows of continental aerosol. The largest black carbon content is a salient feature of the atmosphere of the North and Baltic Seas, surrounded by land: average values of concentrations are 210 ng/m$^3$, and modal values are 75 ng/m$^3$. In other regions (except in the south of the Barents Sea), the average black carbon concentrations are 37–44 ng/m$^3$ (modal concentrations are 18–26 ng/m$^3$). We discuss the specific features of the spatial (latitude-longitude) distributions of black carbon concentrations, relying on ship-based measurements and model calculations (MERRA-2 reanalysis data). A common regularity of the experimental and model spatial distributions of black carbon is that the concentrations decrease in the northern direction and with the growing distance from the continent: from several hundred ng/m$^3$ in the southern part of the North Sea to values below 50 ng/m$^3$ in polar regions of the ocean.

**Keywords:** aerosol; black carbon; spatial distribution; North Atlantic; Arctic Ocean

## 1. Introduction

Atmospheric aerosol plays an important role in the processes of the formation of global and regional climates [1–3]. Increased attention has been devoted in the recent two decades to studying absorbing aerosol (black and brown carbon), which exerts a direct and indirect radiation effect comparable to that due to greenhouse gases [2–4]. In addition, carbon-containing aerosol affects the ecologic state of natural environment and human health [5,6].

Despite the long-term studies, for many regions there remains an uncertainty regarding the emission sources and the transport pathways of absorbing aerosol, the regularities of its variations, and its environmental impact. Remote regions of the ocean and Arctic zone, where the regular measurements are limited or impossible, are the least studied ones.

The content and variations of absorbing aerosol in the atmosphere over the ocean are influenced by ship traffic [7,8]; however, a larger contribution is introduced by outflows

of polluted air from continents, where the main sources of emissions of black and brown carbon are concentrated [9–16], namely: forest fires, industrial plants, transport, heating systems of residential buildings, and associated-gas and biomass combustion. Owing to the atmospheric circulations, black and brown carbon in the composition of submicron aerosol is transported hundreds and thousands of kilometers away to the neighboring areas.

A rough idea of aerosol characteristics over the North Atlantic and the seas of Northern Europe can be gained from the results of multiyear monitoring at island and coastal stations in Greenland, Spitsbergen, Scandinavian Peninsula (see, e.g., [17–19]). However, reliable (actual) data in specific marine regions can be obtained only from measurements onboard research vessels [20–26]. It should be noted that aerosol characteristics exhibit high synoptic-scale variations (about 2 orders of magnitude), associated with a change of air masses. Therefore, short periods of separate marine expeditions are barely enough to provide representative data, characterizing the study region; and longer-term measurements under a variety of atmospheric conditions are required.

Deficient field measurements of aerosol characteristics in high-latitude regions of the ocean are replenished by model calculations, based on an inventory of the sources of aerosol pollutants and statistical data on the trajectories of air mass motion [9–16]. Recent studies started using reanalysis products of aerosol characteristics, based on the measurements of aerosol optical depth, models of meteorological fields, atmospheric circulations, and 3D distributions of different aerosol types [27–29]. However, to verify the results of model calculations, data from actual measurements of aerosol characteristics are also required. This is especially important for high-latitude regions, which have statistically unrepresentative input information, required for the simulation [30].

In 2007–2020 we have conducted almost yearly measurements of black carbon concentrations onboard research vessels (RVs) in the North Atlantic and Arctic Oceans. Black carbon content was measured either using an MDA aethalometer [31], or by means of multi-hour air pumping through filters with a subsequent measurement of the extinction coefficient of the deposited substance, using an absorption photometer [32]. A comparative analysis of results from parallel measurements of the black carbon concentrations by two methods [33] confirmed the agreement of the data and the possibility of their joint use.

The results of determining the black carbon concentrations in different expeditions and by different methods were considered before separately (see, e.g., [23–26]). In this work, we carried out a statistical generalization and analysis of all our measurements of concentrations of black carbon in the composition of atmospheric aerosol over the North Atlantic and the seas washing Northern Europe. Preliminary results from this analysis were discussed in the report [34]. Joint use of two data types made it possible to obtain more reliable statistical characteristics of black carbon concentrations in separate marine regions, and estimate, for the first time, the specific features of their spatial (latitude-longitude) distribution in the study region.

## 2. Characterization of Methods and Expedition Data

The aethalometry method is most widespread for determining the concentrations of absorbing components of aerosol (henceforth black carbon concentration $M_{BC}$, for brevity). The method essentially consists of air pumping through a special filter and of measuring the coefficient of extinction of radiation by the layer of absorbing particles, deposited on a filter. In different practical method implementations, there are differences in the instruments used, measurement regimes, and methods for calculating the black carbon concentrations (see, e.g., [35–37]). However, in all cases there is a common physical basis [38]: the dependence of the measured concentration $M_{BC}$ on variations of the logarithm of the signal of blackening (extinction coefficient) of the filter after a known volume of air is pumped through it or over a specified time of a single measurement cycle.

We utilized two variants of the $M_{BC}$ measurements: (a) using aethalometers of the MDA type [31]; and (b) by the method of aerosol sampling on quartz fiber filters with a subsequent measurement of the extinction coefficient of the filters with absorption pho-

tometer in the wavelength range of 600–700 nm [32]. The MDA's principle of operation is analogous to that of a Hansen aethalometer [38]. The MDA aethalometer was operated in an hourly measurement mode, with one cycle lasting for 10–20 min. The instrument was initially calibrated using black carbon particle generator and simultaneous concentration measurements by aethalometry and gravimetry method [39]. That is, the concentration of absorbing substance was measured in the equivalent of elemental black carbon (EBC mass concentration). To minimize technogenic impacts, the aethalometers were installed in the place most remote from ventilator shafts and ship's funnel, namely: on the upper deck (at the height of 12–15 m above sea level), in frontal part of the vessel. The measurement quality was controlled by an operator: the aethalometer measurements and sampling were stopped in periods of unfavorable weather conditions (spray, drizzle) and explicit technogenic impacts. To eliminate short-term (as short as three-hour) bursts in the measured concentrations, caused by technogenic impacts, automatic data filtering was additionally carried out, using the "three-sigma" statistical criterion. Information on MDA construction, measurement regime, algorithm of data processing, and intercalibrations was given in more detail in [31,40].

In the second method of the $M_{BC}$ measurements, the aerosol samples were collected on the filters for duration from a few hours to a few days. Owing to the long-period measurements, we obtained a linear dependence of the mass concentration of absorbing substance, deposited on the filter, on the extinction coefficient [41]. The extinction coefficients of the filters were measured under the laboratory conditions using an absorption photometer from Institute of Atmospheric Physics, Russian Academy of Sciences (IAP RAS photometer), analogous to the photometer of the Lawrence Berkeley National Laboratory (USA) [41]. The air intake equipment was located next to the ship's command bridges. The measurements used Tissuquartz ™ filters made of pure quartz fibers, without a binder, from Pall Life Sciences (New York, NY, USA), which do not require additional preparation. After exposure, the filters were stored and transported in sealed zip bags in a plastic box. The reliability of the data obtained was verified in 1988 and 1989 through a comparison of extinction coefficients of aerosol samples (15 and 76 filters), using two (IAP RAS and Berkeley) types of the absorption photometers. The intercalibrations [42], carried out together with A.D.A. Hansen, showed a good agreement of measurements by two photometers: the cross-correlation coefficient had been 0.997.

The merits of the aethalometers are their high speed of operation and time resolution: seconds or minutes. However, under the conditions of low black carbon content (such as in the marine or Arctic atmosphere), these advantages are partly lost because of insufficient instrument sensitivity: the smaller the black carbon concentration and time of a single measurement cycle, the smaller the black carbon concentration and the time of a single measurement cycle, the larger the frequency of recording the negative $M_{BC}$ values. Therefore, there is the need to accumulate and average the data over a longer period of time: from tens of minutes to a few hours. That is, the virtual time resolution of aethalometer approaches the sampling period in the filter method.

In measuring the black carbon concentrations in marine expeditions, it is necessary to take into account still another specific feature: the obtained $M_{BC}$ values can be distorted by episodic impacts of smoke from the ship's funnel, pollution from ventilator shafts, and sea spray. For aethalometer measurements, there is the possibility of identifying and sorting out the technogenic bursts; while for multi-hour sampling on filters, this is barely feasible.

In order to justify the joint use of the data from two methods, we compared the measurements of black carbon concentrations, performed by collecting samples on the filters (designated as $M_{BC1}$) and using an MDA aethalometer (designated as $M_{BC2}$). The comparative analysis was carried out using the results from parallel (simultaneous) measurements of the concentrations $M_{BC1}$ and $M_{BC2}$ in 67th and 80th cruises of RV "Akademik Mstislav Keldysh".

In order for the two data types to be compared correctly, the instrumental measurements were referenced to sampling time, and the average concentrations $M_{BC}$ over this

period were calculated. The comparative analysis was carried out using data, satisfying two representativeness requirements: (a) samples should be collected for more than 3 h; and (b) no less than one measurement cycle of the aethalometer should be in the first and second halves of the sampling period. The total amount of data, selected for a comparison, had been 84 joint $M_{BC1}$ and $M_{BC2}$ values.

Figure 1 illustrates the regression of the values "$M_{BC1}$–$M_{BC2}$" and the histogram of the differences $\Delta$; and Table 1 presents the statistical characteristics, reflecting the interrelations between data from two methods of measuring the concentrations of black carbon: cross-correlation coefficients $R$, average values of the differences $\overline{\Delta} = (1/n) \Sigma \Delta_i$, standard deviation of the differences $\sigma$, and the linear regression equations in two variants: (1) $M_{BC1} = a \cdot M_{BC2}$; and (2) $M_{BC1} = b + c\,M_{BC2}$. From the data above it follows that, on the average, the measurements of the concentrations $M_{BC1}$ and $M_{BC2}$ quite well agree.

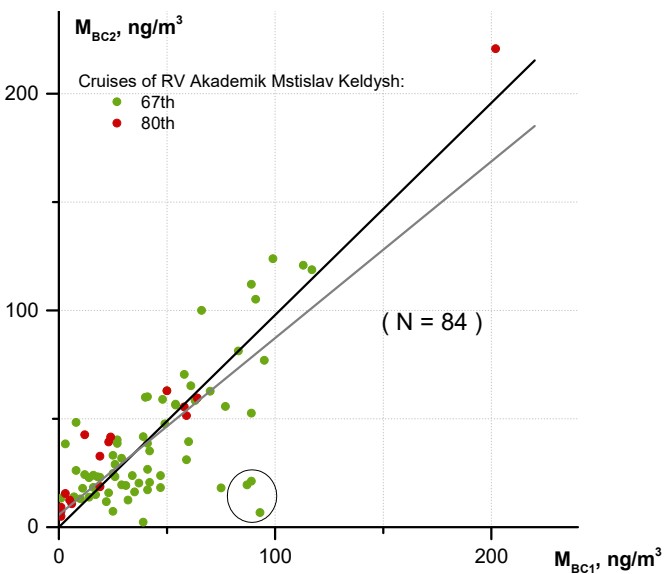

**Figure 1.** Regression relationship of black carbon concentrations measured by two methods.

**Table 1.** Statistical characteristics of the regression relationship $M_{BC1}$–$M_{BC2}$.

|  | $R$ | $\overline{\Delta}$, ng/m$^3$ | $\sigma$, ng/m$^3$ | $a$ | $b$, ng/m$^3$ | $c$ |
|---|---|---|---|---|---|---|
| All data | 0.79 | 2.01 | 21.78 | 0.90 | 5.77 | 0.82 |
| Outlier data | 0.87 | −0.65 | 16.99 | 0.98 | 3.78 | 0.92 |

We turn attention to the presence of a systematic component in the data compared: the $M_{BC1}$ values are 2 ng/m$^3$ larger than $M_{BC2}$ (see first raw in the Table 1). Moreover, there were very large differences $\Delta > 60$ ng/m$^3$ in three cases (circled in Figure 1). A test according to the statistical criterion "three-sigma" showed that the largest $\Delta$ values are bursts due to low-quality measurements by any method. It is these bursts that led to the appearance of the systematic error. After three bursts are eliminated, the characteristics of the regression relationship become better (see the second row in the table): the systematic component decreases to 0.65 ng/m$^3$, and the standard deviation decreases to 17 ng/m$^3$. Thus, the comparisons confirmed the usability of the black carbon concentrations, obtained by different methods. Because the average difference is minor, and the parameter of the linear regression is close to unity ($a = 0.98$), $M_{BC1}$ and $M_{BC2}$ can be used jointly even without applying the correction (intercalibration) coefficients.

Table 2 presents information on 21 expeditions, in which the black carbon concentrations were measured by any (MDA or Samples) method. The routes of the marine expeditions are shown in Figure 2. The dash-dotted line in the figure denotes five areas

for which the statistical characteristics of $M_{BC}$ were calculated: (1) Baltic and North Seas (denoted as BNS); (2) North Atlantic (NA); (3) Norwegian Sea (NS); (4) Barents Sea (BS); and (5) South of the Barents Sea (SBS). We clarify the boundaries of the identified regions: from 60° N in the north and up to 0° E in the west for BNS; from 56° to 66.6° N and up to 0°/−10° W in the east for NA; from 10° W to 20° E and northward of 60° N for NS; from 20° E to Novaya Zemlya and northward of 71° N for BS; and from 68° to 71° N for SBS. The expedition measurements were conducted from May to October. Most (83%) of the data were obtained in the period from July to September (Figure 3).

**Table 2.** Expeditions measuring black carbon concentrations (MDA—instrumental measurements; Samples—sampling for filters).

| № | Period | Expedition Names | MDA | Samples |
|---|---|---|---|---|
| 1 | September–October 2007 | 54th cruise RV *Akademik Mstislav Keldysh* | + | − |
| 2 | September–October 2011 | 59th cruise RV *Akademik Mstislav Keldysh* | − | + |
| 3 | August–September 2013 | NABOS-2013, RV *Akademik Fedorov* | + | − |
| 4–6 | July–October 2015 | 62nd–64th cruises RV *Akademik Mstislav Keldysh* | − | + |
| 7 | August–September 2015 | NABOS-2015, RV *Akademik Tryoshnikov* | + | − |
| 8 | July–August 2016 | 66th cruise RV *Akademik Mstislav Keldysh* | + | − |
| 9 | August–October 2016 | 67th cruise RV *Akademik Mstislav Keldysh* | + | + |
| 10 | July 2017 | RV *Professor Molchanov* | + | − |
| 11–12 | July–August 2017 | 68th, 69th cruises RV *Akademik Mstislav Keldysh* | − | + |
| 13 | June–August 2018 | 71st cruise RV *Akademik Mstislav Keldysh* | + | − |
| 14 | August–October 2018 | 72nd cruise RV *Akademik Mstislav Keldysh* | − | + |
| 15 | August–September 2018 | "Arctic-2018", RV *Akademik Tryoshnikov* | + | − |
| 16–18 | May–September 2019 | 75th–77th cruises RV *Akademik Mstislav Keldysh* | − | + |
| 19 | July–September 2019 | "Transarctic-2019", RV *Professor Multanovsky* | + | − |
| 20 | July–August 2020 | 80th cruise RV *Akademik Mstislav Keldysh* | + | + |
| 21 | August–September 2020 | 81st cruise RV *Akademik Mstislav Keldysh* | − | + |

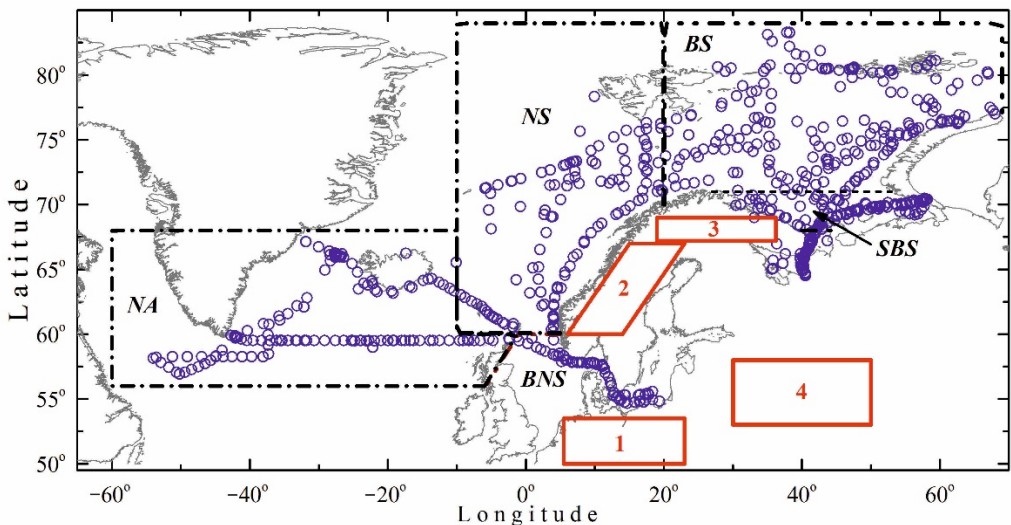

**Figure 2.** Route map of expeditions (measurement sites), boundaries of analyzed sea areas (dash-dotted lines) and test continental areas (red lines).

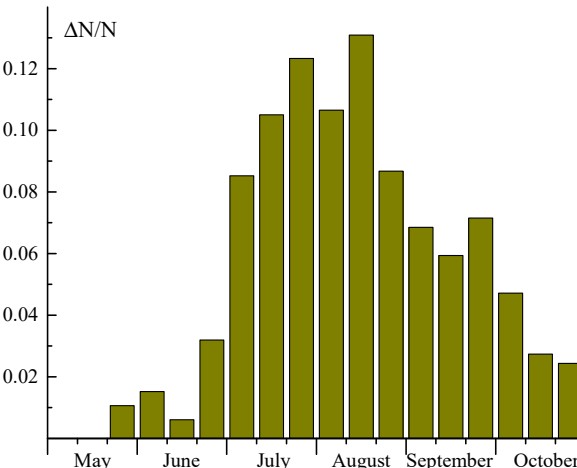

**Figure 3.** Histogram of the distribution of the relative amount of measurement data in different months.

A joint dataset was compiled as follows. The sampling period was mainly 7–9 h. To equalize the statistical weights of two data types, the time scale of instrumental $M_{BC}$ measurements was made closer to the typical sampling period. Namely, hourly aethalometer measurements were used to calculate the average $M_{BC}$ values for three most characteristic sampling periods: 00:00–08:00, 08:00–16:00, and 16:00–24:00 GMT. We also note that the joint dataset was compiled to eliminate the measurements with a short (<3 h) sampling period, which sometimes gives unreliable information. The total amount of data, selected for further analysis, was 657 $M_{BC}$ values (266 days of measurements). Generalization of $M_{BC}$ measurements by two methods made it possible to increase the amount of data and the duration of the total measurement period by about a factor of two. Moreover, distribution of the data over the study regions became more uniform.

## 3. Discussion of the Results

### 3.1. Spatial Distribution of Black Carbon Concentrations

Figure 4 gives a general idea of how black carbon concentrations vary in different marine regions. It can be clearly seen that the concentrations and the $M_{BC}$ variations are larger over the Baltic, North Seas, and in the south of the Barents Sea. Figure 5 presents the average latitude-longitude distribution of black carbon concentrations, plotted using the method of Thin Plate Spline (TPS) interpolation [43]. From the figure it can be seen that the concentrations $M_{BC}$ decrease from ~220 ng/m$^3$ in the southern part of the North and Baltic Seas to the values less than 50 ng/m$^3$ in the northern regions of the ocean. In addition to the tendency of decreasing the concentration $M_{BC}$ with latitude, the factor of continental impact has been well manifested. In particular, owing to its proximity to the Kola Peninsula, the concentrations $M_{BC}$ in the south of the Barents Sea (68°–71° N) turned out to be the same as in the middle part of the North Atlantic at more southern latitudes (<57° N). Evidently, the increased $M_{BC}$ values in coastal regions are explained by their proximity to the continental sources of black carbon emissions in conjunction with the heavier ship traffic. On the whole, the spatial distribution is characterized by decreased concentrations of $M_{BC}$ in the northwestern direction. The gradients of decreasing concentrations with the growing distance from the land in three sectors of the ocean will be considered in more detail in Section 3.3.

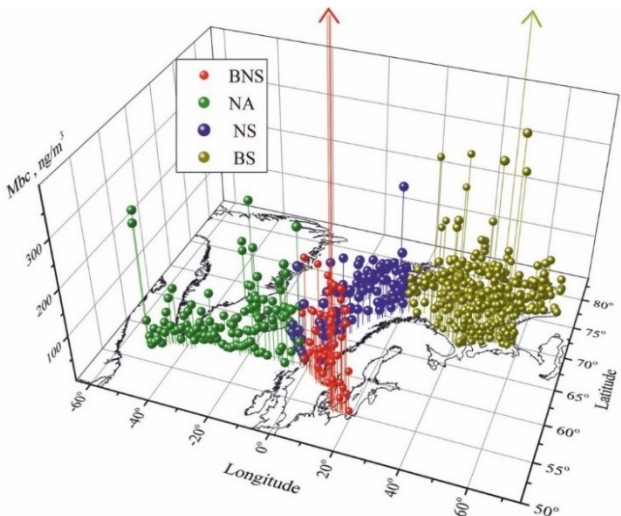

**Figure 4.** Variations in black carbon concentrations in different regions: BNS—Baltic and North Seas; NA—North Atlantic; NS—Norwegian Sea; BS—Barents Sea.

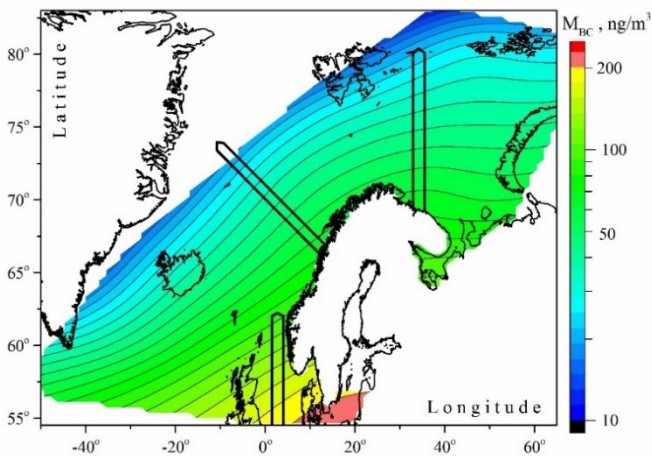

**Figure 5.** Average spatial distribution of black carbon concentrations; arrows indicate the directions of decreasing $M_{BC}$ concentrations in the sectors of the North, Norwegian and Barents Seas, which are analyzed in Section 3.2.

From the frequency histograms (Figure 6a) it can be seen that the most part (>90%) of the $M_{BC}$ values in the total dataset is concentrated in the range up to 120 ng/m³. In nine cases (1.3% of the total number), the black carbon concentrations have values of 300–2500 ng/m³, which are characteristic of continental and urban regions. The black carbon concentrations in excess of 300 ng/m³ were obtained in the water basins of the North and Baltic Seas, surrounded by densely populated and industrially developed European countries, as well as in the Barents Sea.

### 3.2. Statistical Characteristics of Black Carbon Concentrations

The statistical characteristics of $M_{BC}$ were calculated for five regions (see Figure 2), which can differ by outflows of continental aerosol: (1) Baltic and North Seas (BNS); (2) North Atlantic (NA); (3) Norwegian Sea (NS); (4) Barents Sea (BS); and (5) South of the Barents Sea (SBS). Table 3 shows the mean, modal (most probable) values, standard deviations (SD) and coefficients of variation (V, %). From our data it follows that the average $M_{BC}$ values over the North and Baltic Seas (205.6 ng/m³) are a factor of ~5 larger than over the Arctic seas. Modal $M_{BC}$ values are also much larger (Figure 6b): 75 and 18–28 ng/m³, respectively. The right boundary of the $M_{BC}$ frequency plot shifts toward larger values in the

following order: NA–BS–NS–SBS–BNS. In the Arctic zone, the atmosphere in the southern part of the Barents Sea stands out in a higher black carbon content ($M_{BC}$ = 60 ng/m$^3$).

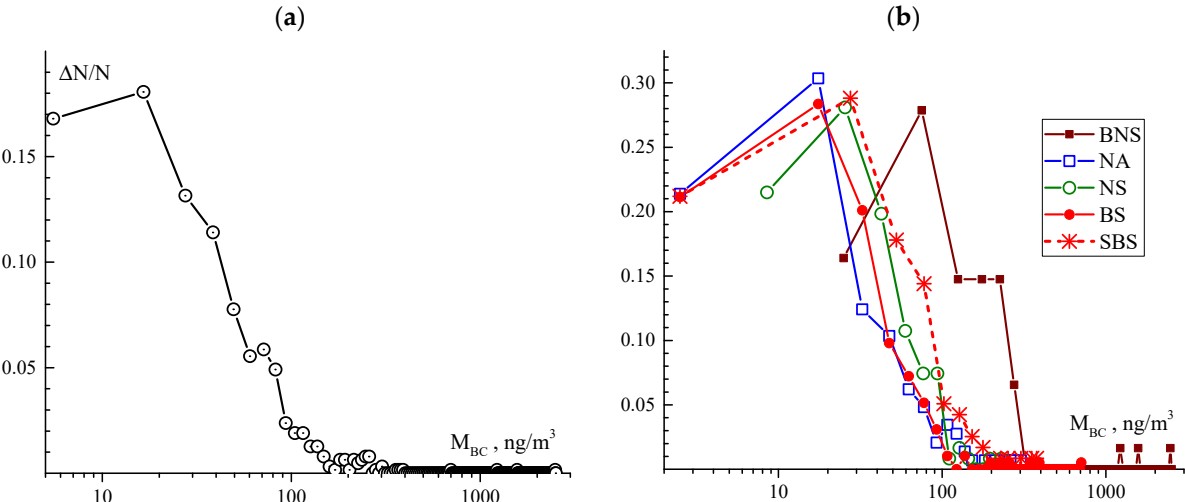

**Figure 6.** Histograms of repeatability of $M_{BC}$ values in the total data set (**a**) and in individual regions (**b**).

**Table 3.** Statistical characteristics of $M_{BC}$ (ng/m$^3$) in different regions.

| Seas (Regions) | Mean ± SD | Mode | V, % | Amount of Data |
|---|---|---|---|---|
| 1. North and Baltic Sea (BNS) | 205.6 ± 383.7 | 75 | 187 | 61 |
| 2. North Atlantic (NA) | 44.1 ± 54.5 | 25 | 123 | 146 |
| 3. Norwegian Sea (NS) | 44.2 ± 36.7 | 26 | 83 | 121 |
| 4. Barents Sea (BS) | 37.2 ± 67.9 | 18 | 182 | 211 |
| 5. South of the Barents Sea (SBS) | 60.0 ± 66.5 | 28 | 111 | 118 |
| Arctic seas (regions 3, 4) | 39.7 ± 58.5 | 26 | 147 | 332 |

It is important to note that the increased $M_{BC}$ variations, caused by the outflows of absorbing aerosol, are observed not only in coastal zones, but also in remote regions of the ocean. As an example, we present two cases when the black carbon concentrations were a factor of three larger than the average level of values for the respective regions: $M_{BC}$ = 90 ng/m$^3$ in the North Atlantic (southward of Greenland) on 11 July 2018; and $M_{BC}$ = 122 ng/m$^3$ in the northeastern part of the Barents Sea on 11 September 2016. Analysis of back trajectories of air mass motion [44] and the centers of temperature anomalies (fires) [45] showed (Figure 7) that the increase in $M_{BC}$ in the first (second) case was impacted by the long-range transport of smokes from forest fires in the north of Canada (on the territory of Siberia).

Thus, the average black carbon content in different marine regions, as well as the $M_{BC}$ variability range, is determined by the frequency and strength of outflows of absorbing aerosol from continental sources. The effect of continental outflows is manifested in two ways. In a relatively small number of cases, the aerosol and black carbon concentrations rapidly increase at the observation site immediately during the outflow, if it is accompanied by a descending air motion to the surface layer. Moreover (in all cases), the impact is indirect and integrated in character: each continental outflow and a set of these entail a growth of the average level of aerosol and black carbon concentrations in the study region, including the measurement site.

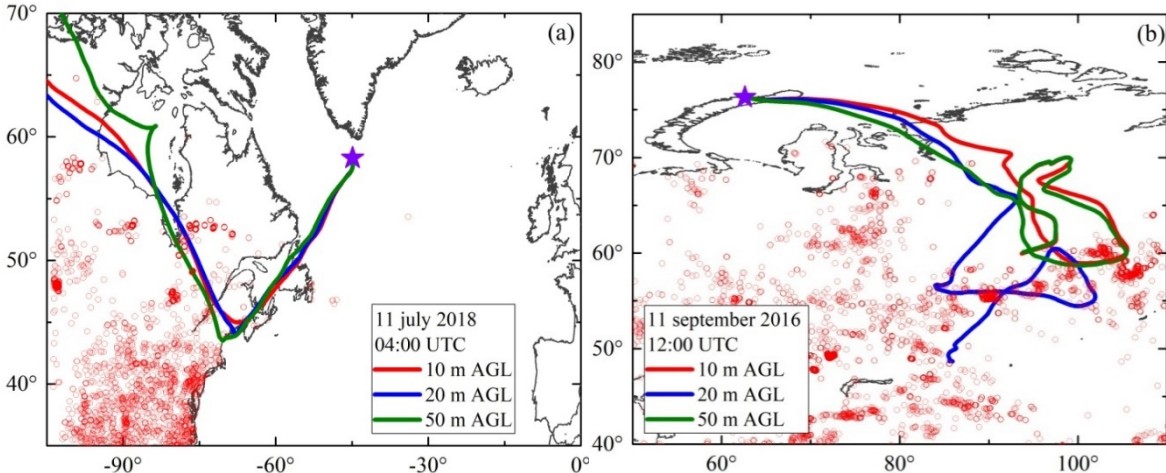

**Figure 7.** Back trajectories of air mass motion to the measurement sites (stars) at heights of 10, 20 and 50 m above ground level (AGL) in the situations on 11 July 2018 in the North Atlantic (**a**) and 11 September 2016 in the northeastern part of the Barents Sea (**b**); red dots indicate fire centers (temperature anomalies).

### 3.3. Comparison of Measurements and MERRA-2 Reanalysis Data

In recent years, the spatiotemporal variations in the black carbon concentrations are increasingly analyzed using the MERRA-2 reanalysis results [27–29], being a product of assimilating the measurements of atmospheric aerosol optical depth (AOD), models of meteorological fields, 3D distributions of different types of aerosol and of air mass transports. For aerosol climatology, the reanalysis data are advantageous (relative to ship measurements) from the viewpoint of their global coverage and spatiotemporal resolution. Therefore, it was interesting to compare the spatial distributions of $M_{BC}$, based on actual $M_{BC}$ measurements in marine expeditions and modeling (MERRA-2) data.

Model distributions were plotted using monthly (July, August, September, October) average $M_{BC}$ values, calculated from multiyear (2007–2020) reanalysis data with the spatial resolution of $0.5° \times 0.625°$ [46]. A preliminary analysis of the data obtained revealed anomalously high monthly average black carbon concentrations during September–October 2014 in the atmosphere of Iceland ($M_{BC} > 1000$ ng/m$^3$), as well as over the ocean within the radius of a few hundred kilometers. This $M_{BC}$ anomaly had been time coincident with the eruption of the Bárðarbunga volcano that began in late August 2014. A characteristic feature of this eruption had been that large amounts of sulfur dioxide gases were emitted [47,48], rather than the absorbing volcanic ash. That is, it is more probable that the anomalous $M_{BC}$ values were due to reanalysis results (assimilation of dataset) being distorted under nonstandard atmospheric conditions. However, we excluded from further analysis the exotic data for September and October 2014.

From Figure 8 it can be seen that the black carbon concentrations in all months decrease in a northern or northwestern direction. In addition to this common regularity, we can clearly discern the difference in the spatial $M_{BC}$ distributions when going from summer (July–August) to the fall (September–October). During summer, the region of relatively high concentrations of $M_{BC}$ (more than 70–100 ng/m$^3$) encompasses a vast territory of ocean up to 70–75° N. During the fall, the contour lines of the concentrations shift in the southeastern direction (toward continent); and in certain regions there are changes in the gradients of decreasing concentrations of $M_{BC}$ with an increasing distance from the land.

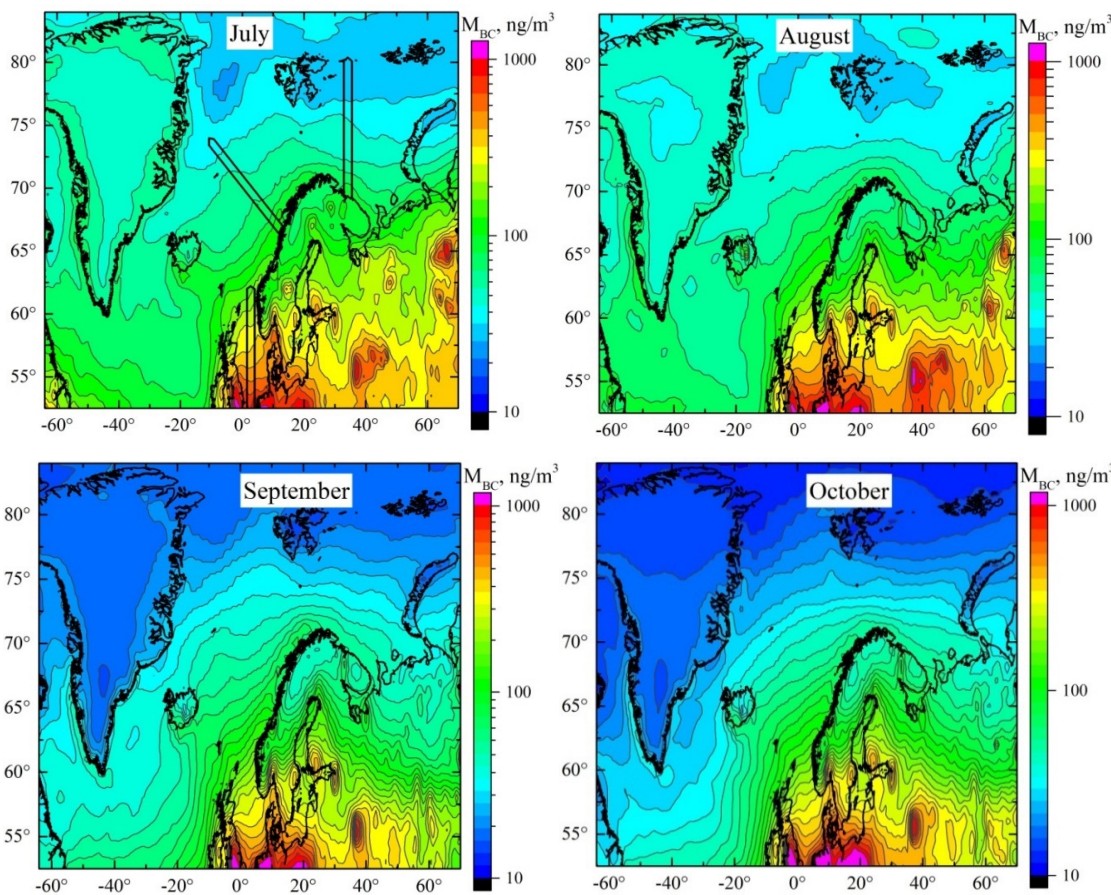

**Figure 8.** Average monthly (July, August, September, October) spatial distributions of $M_{BC}$ according to MERRA-2 reanalysis; black arrows indicate the directions of decreasing concentrations in the sectors of the North, Norwegian and Barents Seas, which are analyzed in Figure 9.

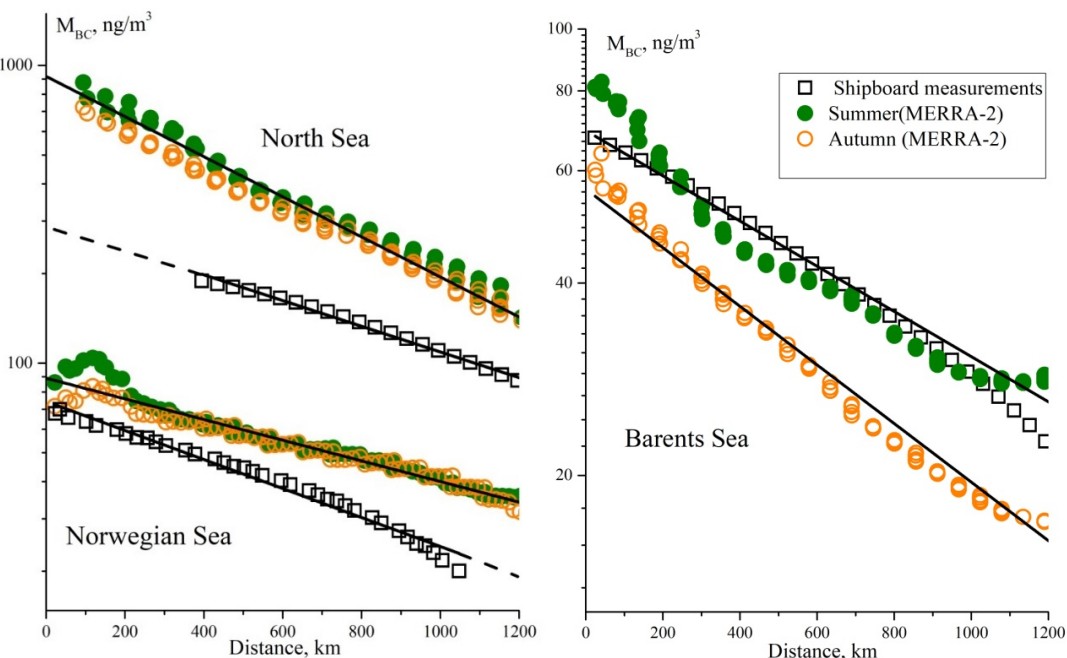

**Figure 9.** Changes in black carbon concentrations with distance from land over the North, Norwegian and Barents seas (lines show extrapolations using an exponential function).

To obtain quantitative estimates, we analyzed changes in the concentrations of $M_{BC}$ in a direction away from the land at a distance of 1200 km in three regions (indicated by black arrows in Figures 5 and 8): (a) over the North Sea in the northern direction; (b) over the Norwegian Sea in the northwestern direction; and (c) over the Barents Sea in the northern direction. Figure 9 shows how the concentrations of $M_{BC}$ change as functions of the distance $L$ for the summer and fall. For comparison purposes, the figure also shows the $M_{BC}$ ($L$) variations, plotted using data from a TPS interpolation of ship-based measurements (Figure 5). The calculations showed that, in all cases, the variations in the concentrations of $M_{BC}$ are well approximated by the exponential dependence: $M_{BC}$ ($L$) = $a \cdot \exp(-b \cdot L)$. The approximation parameters are presented in Table 4. We will consider first the $M_{BC}$ ($L$) dependences, obtained from MERRA-2 reanalysis data.

**Table 4.** Parameters of exponential approximation of the dependence $M_{BC}$ ($L$) in three sea areas.

| Parameter | North Sea | | Norwegian Sea | | Barents Sea | |
|---|---|---|---|---|---|---|
| | **MERRA-2 Reanalysis Data** | | | | | |
| | Summer | Autumn | Summer | Autumn | Summer | Autumn |
| $a$ | 914.2 | 765.1 | 94.6 | 83.6 | 69.4 | 56.1 |
| $b \cdot 10^{-3}$ | 1.50 | 1.40 | 0.86 | 0.74 | 0.81 | 1.07 |
| | **Shipborne measurements of $M_{BC}$** | | | | | |
| $a$ | 294.4 | | 73.7 | | 72.2 | |
| $b \cdot 10^{-3}$ | 0.99 | | 1.12 | | 0.92 | |

From the data presented, it follows that the sector of the North Sea stands out in a higher gradient of the $M_{BC}$ variations: the exponent $b$ is a factor of 1.5–2 larger than in other regions. Over the Barents Sea, we clearly discern the seasonal differences: in autumn, the concentrations at all distances from the land became 25–60% smaller, and the gradient of the $M_{BC}$ ($L$) variations became a little stronger. In two other regions, the dependences $M_{BC}$ ($L$) in both seasons almost coincide (a common approximation curve is presented).

The question arises as to why the spatial $M_{BC}$ distribution is transformed from the summer toward the fall (see Figure 8): is this due to seasonal decrease in the black carbon content in the atmosphere of continental regions, or to decreased air transports from the territory of Europe in the direction of ocean? Clarification of the causes for the spatial $M_{BC}$ variations over the ocean requires a separate study, which is beyond the scope of this work. Therefore, we confined ourselves to estimates of the seasonal variations in one of the possible predictors, i.e., the black carbon concentrations in continental regions (indicated by red color in Figure 2). Most evident is the marine atmosphere, which is influenced by three coastal regions: (1) north of continental Europe, (2) Scandinavian Peninsula, and (3) Kola Peninsula. Additionally, we considered a region remote from the ocean, i.e., Region 4 (Moscow agglomeration), with a relatively high black carbon content.

From Figure 10 it can be seen that the black carbon content in Region 3 decreases from July to October by ~60%, entailing a decrease in the concentrations of $M_{BC}$ over the Barents Sea as well (Figure 9). The black carbon content in Region 2 also decreases (by ~25%), but this barely affected the concentrations of $M_{BC}$ over the Norwegian Sea. Therefore, the decrease in the black carbon emissions in Region 2 was compensated for by more frequent outflows of continental aerosol to the sector of the Norwegian Sea.

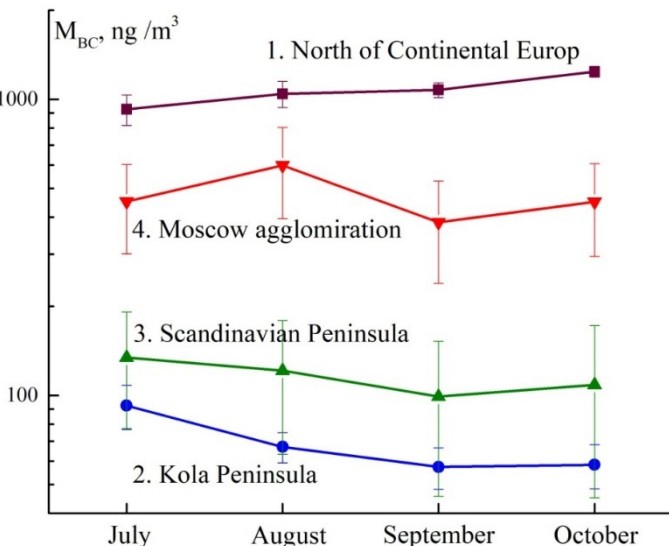

**Figure 10.** Variability of monthly average concentrations of black carbon in four continental regions (MERRA-2 reanalysis data).

An August maximum appeared in the seasonal behavior of the black carbon concentrations in Region 4 due to massive forest fires in 2010 [49]. However, (either with or without exclusion of this anomalous year), in this case there is again a tendency of decreasing black carbon content from summer toward the fall, favoring, to some degree, the decrease in the $M_{BC}$ concentrations over the ocean.

The north of the continental Europe (Region 1) exhibits an opposite regularity: the black carbon content increases by about a third from July to October. However, this yielded no increase in the concentrations of $M_{BC}$ over the North Sea: they are a little larger during summer than the fall (see Figure 9). This behavior can be explained by the offsetting effect of the circulation factor, i.e., by the decreased number of air outflows from the direction of the land. Thus, the seasonal (summer–fall) change in the spatial $M_{BC}$ distribution in the European sector of the Arctic Ocean was influenced by both factors: the decrease of the black carbon content in most land regions (2–4) and the change in the transports of continental aerosol.

Maps of the spatial $M_{BC}$ distributions from reanalysis data (Figure 8) qualitatively agree with data from experimental measurements (Figure 5). In both cases, the black carbon concentrations decrease in the northern or northwestern direction: the average $M_{BC}$ values are a few hundreds of ng/m$^3$ in the southern part of BNS, and they are a few tens of ng/m$^3$ in high-latitude regions of the Arctic Ocean. We point to the quantitative differences between model and experimental dependences $M_{BC}$ ($L$), presented in Figure 9. The experimental $M_{BC}$ ($L$) values lie below model values in the North and the Norwegian Seas, while in the Barents Sea they almost coincide with the latter in summer period. The exponent $b$, which characterizes the gradient of the $M_{BC}$ ($L$) variations (see Table 4), is close to unity in experimental dependences over all seas. A significant difference in the exponent $b$ between model and experimental dependences of $M_{BC}$ ($L$) was manifested only in the North Sea in the summer period. We note that the ship-based $M_{BC}$ measurements were carried out only in the northeastern part of the North Sea. Accordingly, the dependences of $M_{BC}$ ($L$), obtained using data from TPS interpolation of experimental data, are approximate in character.

The differences between experimental and model $M_{BC}$ distributions, considered here (Figures 5, 8 and 9), were partly due to different spatiotemporal averaging scales and different amounts of the two data types. Maps of the model $M_{BC}$ distributions are the result of an averaging of a large amount of data over the entire period (2007–2020), while ship-based measurements were carried out only at specific locations and in specific periods

of each expedition. Some distortions could also be introduced by a TPS interpolation of ship-based data, presented in Figure 5. Therefore, to obtain correct estimates of the quantitative differences, we compared model and experimental $M_{BC}$ values matched to be coincident in time (within $\pm 4$ h) and collocated in coordinates (within $\pm 1°$).

Figure 11a illustrates the regression between matched $M_{BC}$ values, based on the model (MERRA-2) calculations and measurements onboard the research vessel (RV). On average, the concentrations of $M_{BC}$ from reanalysis data are larger than from actual measurements. It can be clearly seen that three data points with very high $M_{BC}$(RV) values (indicated by red color) deviate from the common regularity. Depending on whether these points are accounted for or not, the average discrepancy of the data $\overline{\Delta} = [M_{BC}(\text{MERRA}) - M_{BC}(\text{RV})]$ and the correlation coefficients R strongly differ: $\overline{\Delta} = 26$ ng/m$^3$ and R = 0.5 for the total dataset; and $\overline{\Delta} = 18$ ng/m$^3$ and R = 0.28 after the three points are removed. Overestimated values $M_{BC}$(RV) > 1000 ng/m$^3$ were obtained in the North Sea basin. The causes for large $M_{BC}$(RV) values in this region are difficult to determine unambiguously: either local smoke plumes from oil derricks, or from passing ships, or from industrial plants, located on the coast. Nevertheless, the results of such measurements can hardly be used for a comparison with large-scale ($2° \times 2°$) reanalysis data, in which the local effects are smoothed out.

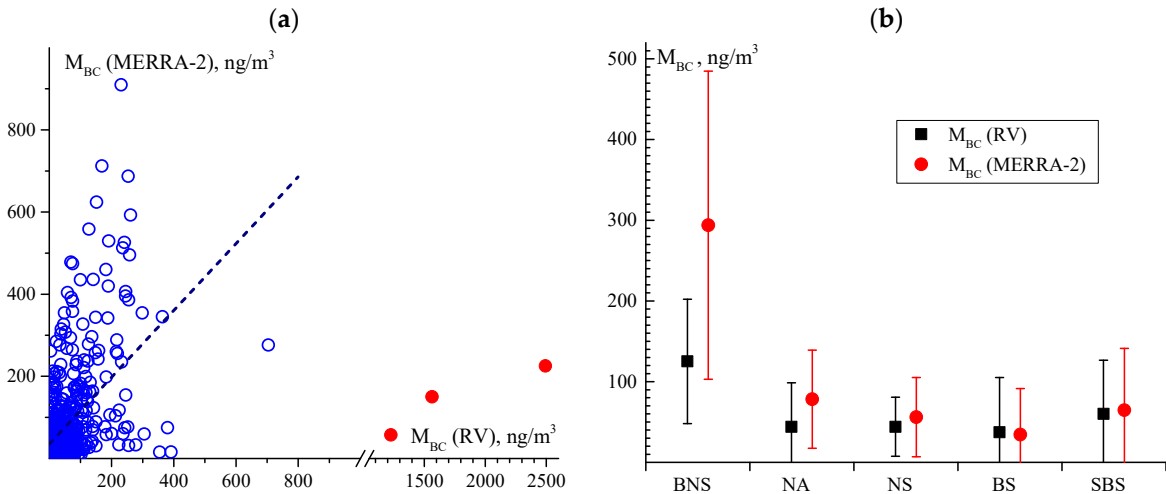

**Figure 11.** (**a**) Regression relationship of $M_{BC}$ concentrations according to modeling (MERRA-2) and ship measurements (RV); (**b**) comparison of black carbon concentrations in five areas (concentration $M_{BC}$ (RV) in the BNS area is given without taking into account three outliers).

Figure 11b compares the average concentrations of $M_{BC}$, calculated for separate regions. An acceptable agreement of model and experimental data is observed in Arctic regions: the relative discrepancy $\delta = \overline{\Delta}/[M_{BC}(\text{RV})]$ is 27% in the Norwegian Sea, $-7\%$ in the Barents Sea, and 8% in the south of the Barents Sea. The two data types differ much more strongly at midlatitudes: $\delta = 43\%$ (or 135%) in the North and Baltic Seas; and $\delta = 78\%$ in the North Atlantic.

Thus, the concentrations of $M_{BC}$ in most regions, on the average, are larger when determined from reanalysis data than from actual measurements. The discrepancy could be due to different spatial averaging scales: the experimental data are the average local $M_{BC}$ values on the track of the vessel over 8 h of measurements; while the reanalysis data were averaged for the same period of time within $2°$ in latitude and longitude. However, the main cause for the difference of $M_{BC}$ (MERRA) from $M_{BC}$ (RV) is the absence or scarcity of (ground-based or satellite) measurements of the atmospheric AOD over the ocean, which are required for the model calculations. The same cause was indicated by the authors of work [30] who analogously compared $M_{BC}$ (MERRA-2) with $M_{BC}$ measurements at four polar stations.

## 4. Conclusions

Results from multiyear measurements of the black carbon concentrations in the atmosphere of the North Atlantic and the European sector of the Arctic Ocean (266 days of measurements in 21 marine expeditions during 2007–2020) were statistically generalized. A comparison of black carbon concentrations from parallel measurements during two expeditions by the filter method and using aethalometer, showed an acceptable agreement of the data (correlation coefficient of 0.87 and standard deviation of 17 ng/m$^3$) and the possibility of their joint use.

It is noted that the spatial distribution of black carbon over the ocean is formed under the influence of outflows of air masses from the direction of continents, where the main emission sources of absorbing aerosol are concentrated. The latitude–longitude distribution of black carbon in this region is characterized by the average concentrations $M_{BC}$ decreasing in the north and northwestern direction: from ~220 ng/m$^3$ in the south of the North Sea to the values below 50 ng/m$^3$ in high-latitude regions of the ocean.

Based on the data obtained, we calculated the statistical characteristics of the black carbon concentrations for five regions, in which the outflows of continental aerosol are differently manifested. Maximal black carbon content is the salient feature of the atmosphere of the Baltic and North Seas, surrounded by the land: the average $M_{BC}$ values are 205.6 ng/m$^3$, modal values are 75 ng/m$^3$, and the variation coefficients are 187%. The relatively high concentrations of $M_{BC}$ are also observed in the southern part of the Barents Sea (68°–71° N): the average concentrations are 60 ng/m$^3$ and the modal values are 28 ng/m$^3$. The black carbon concentrations in the atmosphere of other regions (NA, NS, BS) are comparable in value: 37–44 ng/m$^3$ (modal values are from 18 to 26 ng/m$^3$).

The spatial $M_{BC}$ distribution, plotted using ship-based measurements, is qualitatively similar in shape to the model distributions, calculated using MERRA-2 reanalysis data. Comparison of matched (coincident in time and collocated in coordinates) ship-based and model $M_{BC}$ values showed an acceptable agreement of the data in the atmosphere of the Norwegian and Barents Seas: the relative discrepancy $\delta = \overline{\Delta}/[M_{BC}(\text{RV})]$ is in the range of −7% to 27%. The two data types strongly differ at midlatitudes: $\delta$ = 43% (or 135%) for the North and Baltic Seas; and $\delta$ = 78% in the North Atlantic. On average (for the total dataset), the model-based concentrations are 18–26 ng/m$^3$ larger than those from ship-based measurements.

In the reanalysis data we discerned the seasonal transformation of spatial black carbon distributions in going from summer to fall: the concentrations $M_{BC}$ exceed 70–100 ng/m$^3$ on the vast territory (up to 70–75° N) during summer; in the fall, the concentration contour lines shift in the southeastern direction (toward continent). The black carbon concentrations in the study regions of the ocean decrease exponentially with growing distance from the land. The concentrations decrease most strongly over the North Sea. In the sectors of the Norwegian and North Seas the dependences $M_{BC}$ (L) almost coincide in summer and fall. Over the Barents Sea, the black carbon concentrations decrease by 20–60% during fall; and the gradient of the $M_{BC}$ (L) variations slightly increases.

**Author Contributions:** Conceptualization and writing original draft, S.M.S.; Organization of expeditionary measurements—V.P.S. and A.N.N.; Expeditionary measurements—A.N.N., V.M.K. and V.V.P.; Physical analysis of samples, V.M.K.; Processing, analysis and interpretation of data—D.M.K., V.M.K., A.N.N., Y.S.T. and V.V.P.; Data processing of trajectory analysis of the movement of air masses and MERRA-2 reanalysis—D.M.K. and I.A.K.; Review and editing—S.M.S. and V.P.S. All authors have read and agreed to the published version of the manuscript.

**Funding:** This work was supported by the Russian Science Foundation (Grant No. 21-77-20025).

**Institutional Review Board Statement:** Not applicable.

**Informed Consent Statement:** Not applicable.

**Data Availability Statement:** Data can be obtained from Dmitry M. Kabanov upon request.

**Acknowledgments:** The authors thank their colleagues who participated in measurements and in preparation of instrumentation, i.e., O.N. Izosimova, A.O. Pochufarov, Vas. V. Pol'kin, V.P. Shmargunov, S.A. Terpugova, and P.N. Zenkova. We also thank the organizers of the sites https://giovanni.gsfc.nasa.gov/giovanni (accessed on 10 May 2021), https://ready.arl.noaa.gov/HYSPLIT.php (accessed on 15 April 2021) and https://firms.modaps.eosdis.nasa.gov (accessed on 15 April 2021) for the opportunity to use important information.

**Conflicts of Interest:** The authors declare no conflict of interest.

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
