# Peer review of "Spatial Distribution of Black Carbon Concentrations in the Atmosphere of the North Atlantic and the European Sector of the Arctic Ocean"

_atmosphere, doi:10.3390/atmos12080949_

Round 1

Reviewer 1 Report

See attached file for review.

Reviewer 2 Report

Sakerin et al. present data collected from a composite of observations from various studies conducted on ships in the Northern Atlantic, near northern Europe, and the Arctic Ocean. They describe how they combined data from different observations to create a long term dataset of black carbon in this region, and compared against model calculations of reanalyzed data. They found that black carbon was generally highest near coast and decreased as the observations increased from distance of continental sources.

The study provides important insight about combining observations from multiple studies, which are necessary to try to stitch together long term trends and how aerosol behaves in remote locations that are generally not well observed. The study would be of value to Atmosphere after the authors address the following comments.

(1) General order of paper. The necessary parts are here for the paper to be complete; however, the order that the material is presented sometimes makes it confusing or makes it hard to understand the findings. This includes:
(1a) Description of the regions (line 236-238). These abbreviations are not included in the captions of Fig. 2 and Fig. 11. As this the description of the methods and region, I would recommend moving this line to the paragraph starting at line 173.

(1b) Fig. 11. This goes along with the comment (2) (see that comment for more detail). I would recommend moving this figure and the associated text/analysis to before the description of Fig. 8 to better understand the comparison of model and observations.

(2) Right now, the comparison of observations and model results, specifically the analysis for Fig. 8, is confounded by the fact that the observations, including Fig. 5, is the data from all months while Fig. 8 shows potential differences by months. Have the authors looked at if they look at data in a similar way they evaluated Fig. 8, the gradients change? Right now, it is unclear if the authors can say anything about the gradients in Fig. 8 are real or not, and that the model should be just averaged all together like the data to make a better comparison between the two.

(3) Fig. 7. There is inherent uncertainty and spread for backtrajectories, especially for trajectories that go back for many days. It would be of use to see what the spread in backtrajectories for these two case studies is, and if they all are generally in the region of high fire centers.

(4) The authors have generally done a good job with their statistical analysis. Table 4 would be good for similar analysis (what is the standard deviation, and does that make the values statistically different or not?).

(5) Line 397. I was surprised that removing the data the authors considered to be outliers led to a lower R value. I would have thought the R value would have increased. Can the authors provide more explanation behind this?

(6) Fig. 11. Have the authors tried averaging the observations to be the same spatial scale as the modeled results. Eg, instead of doing data within X distance and doing 1:1 comparison, average the data into the grid box size of the model? Or is that what is being done with Fig. 11a already? It is not clear currently. Fig. 11b seems to be a more sound way to do this type of analysis to minimize potential plumes that impact the scatter plot.

(7) One thing that seems slightly missing in this analysis that would make this paper stronger is what is driving the decreasing black carbon (Fig. 9)? Is it mainly dilution or is it due to deposition/aging? The latter is an ongoing question, especially for remote regions (e.g., Katich et al., JGR, 2018, https://doi.org/10.1029/2018JD029206; Lund et al., 2018, https://doi.org/10.1038/s41612-018-0040-x; Schwarz et al., 2013, GRL, https://doi.org/10.1002/2013GL057775; Schwarz et al., 2017, GRL, https://doi.org/10.1002/2016GL071241). Providing what the values derived for Fig. 9 would provide great new information.

Minor

(1) Line 235, I think the reference is for Table 3 instead of 2, though unclear.

(2) Fig. 5 & 8. It would be good to say in the caption what those arrows represent.

Reviewer 3 Report

The article entitled “Spatial distribution of black carbon concentrations in the atmosphere of the North Atlantic and the European sector of the Arctic Ocean” by Sakerin et al., describes the distribution of black carbon concentrations over northern European arctic regions by measurements performed over years of ship-based investigations. The article is well written and is in line with previously published material with the evaluation and interpretation of the data recorded over 13 years of measurements with research vessels in the Northern European seas. The previous articles have been published (7 articles from the bibliography) already in various journals and all of them are based on the interpretation of the data from impressive research seas campaigns series over the north of Europe.

Measurements performed at the research vessels are limited to the data recorded over the trip periods and could be in the range of 2 orders of magnitude different. For such studies, the main problem is the representative data recorded. However, many campaigns over years of studies indicate interesting trends of the BC concentrations.

How do you treat the interferences from ship emissions? Aerosol concentration could be easily affected by engine emissions, fuel burning, daily living on the ship emissions, etc. Please explain in the text.

How do you rely on the data when the weather is unfavorable? Air mass dynamics could affect the measurements. Please describe the placement of the measurement station on the ship. It is on the side of the ship, front or back? How the sampling is stopped when the engine plume from the back is moved to the front due to the wind change direction. Did the sampling stop when the meteorological wind direction measurements provide such behavior? The data are collected during ship movement or also when the ship is stopped? Do you have front and back ship simultaneous measurements for comparison? Or any replicates of the sampling site? Even a parallel measurement of the 1 m distance sampling could generate different values.

Except for expeditions 9th and 20th, no others collected data by both instrumental and filters. Why do not collect samples by both methods in the campaigns? Or the data were not included in the study? Why?

Please explain how the filters are prepared before the sampling, how the storage of the filters has been performed. What has been the time difference between the filters analysis and filter collection?

Please explain in detail the figure 3. What is delta N/N? If there you use concentration why not represent besides delta N /N also the delta mass/ mass? The numbers are nothing if there is not a distribution of particles on the aerodynamic diameter.

Please add more details about figure 5. There are representations of Mbc gradients from coast to deep in the remote sea but there is some unexpected and strange representation since BS arrow starts from unexpected green coast close to Murmansk, Baltic sea and gulf to Oulu is also unexpected green, there is no indication of heavy pollution from Murmansk, there is some indication that the black carbon comes from central Europe only, no indication of preponderant wind directions from Russia to Europe or any other direction to explain at least the indications on figure 5, there is also no information about the data collected from other territories than Russians, (do you have data in figure 5 from some pollution sources from Sweden and Finland to explain why in the Gulf of Bothnia the measurements indicate green levels for Mbc?).

Figure 6 should be clearly explained. What the data point represents? What measurement do authors count? It is a (dN/N) / Mbc(ng/m3) but for which measurement? If there is just a statistic value independent of the sample source should be mentioned. Also in the text of the article is nothing mentioned about figure 6b.

Line 235 -  there are you talking about table 2?

Section 3.2 Please indicate the temporal relationship between the fires events and measurement samples collection time. The amplitude of the fire events should be also specified. To generate an influence of fires event trajectory and the long-range effect, the fire event size, meteorological conditions, event time, etc should be presented.

Round 2

Reviewer 2 Report

Thank you for your responses. The authors were thorough in responding to both reviewers comments, and it has improved the paper for publication in Atmosphere.

Author Response

We thank the reviewer for his careful reading and helpful comments.

Reviewer 3 Report

The article entitled “Spatial distribution of black carbon concentrations in the atmosphere of the North Atlantic and the European sector of the Arctic Ocean” by Sakerin et al., has been amended by the authors according to the reviewer suggestion and comments and deserves publication with minor changes

Minor revisions:

Abstract:

Please revise the sentence “The black carbon concentrations were measured by aethalometer and filter method.” Since there you consider an instrument and a method. I suggest adding a “The black carbon concentrations were measured by optical and filter method”

Please replace “over ocean” with “over the ocean”. Could consider this over the entire text by adding the article...

2

Please replace “the concentrations of absorbing component” with “the concentrations of absorbing components”.

Please replace “though” with “through”.

Page 10 first paragraph: replace “1200 km” with “of 1200 km”

Author Response

We thank dear Reviewer for helpful comments.

 1. A juxtaposition of the optical (aethalometer) and filter (sampling) methods seems to be inappropriate to us. The point is that the “filter method” (after collecting samples on filters) also employs the optical method: the determination of the extinction coefficients and the concentrations, using an absorption photometer (see Section 2 for more detail). Therefore, we suggest the following alternative:

“The comparison of the two variants of the measurements of the black carbon concentrations showed that the data acceptably well agree and can be used jointly ”

2. We have corrected in the text.

(a) “over ocean” was corrected to “over the ocean” throughout the manuscript.

(b) “concentrations of absorbing components” was corrected to “concentrations of absorbing components”

(c) “though” was corrected to “through”

(d) “1200 km” was corrected to “of 1200 km”